# In-Service and Pre-Service Science Teachers' Enacted Pedagogical Content Knowledge about the Particulate Nature of Matter

Anastasia Buma [1] and Doras Sibanda [2],*

1   School of Education, University of The Witwatersrand, Johannesburg 2000, South Africa
2   School of Education, University of KwaZulu-Natal, Pietermaritzburg 3201, South Africa
*   Correspondence: sibandad@ukzn.ac.za

**Abstract:** The particulate nature of matter is a fundamental concept in science that students in lower grades find difficult to understand. Pedagogical content knowledge (PCK) has been identified as germane for addressing difficult topics and it enhances effective learning. The purpose of this study was to capture the quality of the enacted PCK that practising and pre-service teachers activate during planning. Data were collected through a validated PCK questionnaire which was completed by both practising and pre-service teachers. A rubric was used to code their responses. A Rasch analysis model was used to analyse the five components of the topic-specific PCK construct. Data from an item analysis show that pre-service teachers found the test items to be less difficult than did the practising teachers. We found that there was a statistically significant difference between the two groups of teachers in terms of knowledge activated during planning. These findings show that, in transforming the topic content and concepts of the particulate nature of matter, the pre-service teachers integrated more components of enacted PCK compared to practising teachers. Discussions around the curriculum for both groups of teachers might provide insight into the design of future teacher development programmes.

**Keywords:** pedagogical content knowledge; planning; particulate nature of matter; practising teachers; in-service teachers





## 1. Introduction

Research indicates that knowledgeable and skilful science teachers use their extensive teacher knowledge to transform content when planning a lesson to promote effective learning. Debates on the nature of teacher knowledge for teaching have been ongoing since the seminal work of Shulman in the mid-1980s, which introduced the idea of pedagogical content knowledge (PCK) as an important category, among others, of teacher knowledge used to guide the pedagogical reasoning employed by teachers in transforming content knowledge of a lesson for effective learning [1,2]. In science education, PCK has been widely accepted as one of the leading constructs needed to understand teacher knowledge and enhance its role in supporting effective teaching and learning [3,4]. Different studies have been reported that highlight the different elements of PCK (including, for example, knowledge of learner misconception, representation, key and difficult ideas about a topic, etc.) that can be integrated when transforming topic content to promote student understanding [4,5]. Recently, PCK in science education has been conceptualised as having three forms that locate it within a discipline, topic, and concept [6]. These forms are, firstly, collective PCK (cPCK) that depicts the specialised professional knowledge constructed by a group of subject experts about the concepts and topics in a specific discipline; second is the personal PCK (pPCK), which is the specialised teacher knowledge developed by a teacher from experiences in practice; and lastly, enacted PCK (ePCK), which is the manifestation of the collective PCK and personal PCK in action [6] during planning, actual teaching, and

reflection on the taught lesson. The implication of this model is that teacher education and professional development programmes need not only to impart conceptual understandings about PCK, but also need to create practical opportunities to facilitate the development of the construct. This paper, therefore, seeks to enhance our understanding of the nature of teacher professional knowledge manifested by both practising and pre-service teachers in their planning.

There is evidence suggesting that opportunities for teacher knowledge integration and subsequent knowledge development are afforded by experiences in microteaching [7,8] and actual teaching, during professional development interventions that also include opportunities for lesson study of video-taped lessons [9] and lesson planning and reflection. In this study, practising teachers (PTs) were already teaching science subjects at schools, while the pre-service teachers (PSTs) were registered in science methods courses that afforded peer teaching and microteaching opportunities that promoted development of their enacted PCK about the topic and related concepts. In this study, it is expected that the enacted PCK of both teacher groups would continue to develop during lesson planning, enactment, reflection, and in engaging with peers and others who add value to the education system. The teaching and learning of the particulate nature of matter (PNM) in junior secondary science was labelled as abstract and difficult over three decades ago [10,11] and it remains as such to date [12]. Studies that explore the nature of teacher knowledge at a junior secondary school level have tended to focus more on collective PCK and personal PCK than on enacted PCK [13]. Little research has been reported in science education literature that captures PCK activated by both practising and pre-service teachers during planning [9,14]. Moreover, planning to teach has been branded as a pertinent pedagogical endeavour owing to its benefits in aiding the reflection on and improvement of the PCK required to transform the content of lesson topics for sense-making to students [15,16]. There is common assumption that pre-service teachers have little or no PCK [17]. In this study, we wanted to capture the nature of PCK used in planning to teach PNM by both pre-service and in-service teachers. The research questions that this study addresses are as follows.

What type and quality of enacted PCK is activated when pre-service and in-service teachers plan to teach the particulate nature of matter?

What are the differences in the enacted PCK activated by pre-service and in-service teachers when planning to teach the particulate nature of matter?

## 2. Literature Review

In the country where this study was conducted, PNM is a topic in "Matter and Materials" prescribed as a strand for junior science from Grades 7 to 9 by the Department of Basic Education's Curriculum and Assessment Policy Statement [18]. The curriculum document provides guidelines of the sequence of concepts and the time frame to be spent teaching each concept [18]. In this study, we focused on the PNM topic as prescribed for Grade 8 [18]. The topic was selected because it is part of the basic chemistry taught in lower grades, upper secondary and tertiary education. This topic serves as the basis for understanding key topics such as stoichiometry, chemical bonding, and chemical kinetics. The PNM concept also represents the chemistry content in three representations (microscopic, sub-microscopic, and symbolic). Research has shown that if students do not understand the particulate nature topic from the early years of schooling, they often have difficulties learning the chemistry content thereafter [12]. Boz [19] found that after the teaching of the topic, middle and high school students had difficulties in applying the PNM to explain phase changes. More importantly, the PNM is regarded as one of the most difficult topics for teachers and students [20,21] and has been described as abstract and hard to learn [12]. This view is illustrated by the types of common-sense conceptions held on the topic by students. Among middle and high school students, Ozalp and Kahveci [22] identified some examples of common-sense conceptions that students held: because ice is a solid, its particles are solid, water is a liquid and so its molecules are liquid; individual

atoms can heat and melt and their volume increases. The students also held views that water molecules break up into oxygen and hydrogen atoms when "water evaporates". The high prevalence of common-sense conceptions held by learners on the PNM might suggest that teachers also hold the same common-sense conceptions. Studies on students and teachers [21,22] have shown problems in conceptual understanding of the PNM among both in-service (that is, practising) and pre-service science teachers. It is, therefore, important that both in-service and pre-service natural science teachers demonstrate a strong PCK in teaching the PNM; hence, the need to capture in-service and pre-service junior science teachers' conceptual understanding of the PNM and the teacher knowledge required to address the common-sense conceptions merits further exploration.

### 2.1. PCK Development Involving Quality in PCK Component Interactions

Research has shown that classroom experience of teachers is an important factor in the development of PCK [1,6]. Such experience typically involves planning, teaching, and post-teaching reflection. The development of PCK of novice and experienced teachers has been reported to be different. Certain researchers, for example, Lee and Luft [17] and Krepf et al. [9] reported that expert science teachers have a richer understanding of PCK components as well as of their meaningful blend into a well-organised conceptualisation of PCK compared to novice science teachers. It can be argued that even in-service teachers who have experience in the art of teaching but do not consciously reason pedagogically on lessons will display fragmented PCK and be less likely to help their learners to learn effectively. Debates on how teachers develop PCK are ongoing and at present, there is no single way of describing how teachers develop it. Hashweh [23] acknowledged that PCK develops when a teacher constructs new ways of explaining new or difficult concepts, or, more so, when they draw on experiences in teaching [24]. In a different perspective, it could entail interaction of the teacher's understanding of PCK components with the new content, executed in such a way that the more components used in explaining a concept, the richer and more meaningful will be the teaching and learning of the concept—a phenomenon that has been linked to the quality of the enacted PCK (Miheso & Mavhunga, 2020). Equally articulated is the idea that PCK development is achieved when teachers engage in microteaching [7,8], and during professional development interventions [9]. Sanders, Borko, and Lockard [25] investigated three experienced teachers' knowledge on unfamiliar topics. They found that teachers' insufficient subject matter knowledge prevented their PCK development, and they were not able to transfer expert knowledge. However, Rollnick [26] observed that seven teachers' subject matter knowledge changed when enrolled in a programme. The findings of Rollnick's study showed that teachers acquired new subject matter knowledge and their PCK also developed. Similarly, Aydin and Boz [27] investigated PCK component interactions based on the Pentagon PCK model proposed by Park and Oliver [28] among two experienced chemistry teachers teaching redox reactions and electrochemical cells. Their study revealed that more coherent component integration was observed in both teachers' teaching of the electrochemical cells topic compared to redux reactions, and, more so, the knowledge of learner misconceptions and instructional strategy components were central in the component integrations. Although knowledge of assessment and curriculum were less effective in shaping the teachers' teaching, a further glaring finding was that the integrations were specific to the taught topic.

### 2.2. Planning for Teaching Knowledge

A key component of the knowledge needed by teachers in their practice, according to Ball, Knobloch and Hoop [29], is planning. The planning of a lesson is an important experience for teachers and many initial teacher training programmes have lesson planning as a key component [23]. The planning of a lesson plays an important role in how a teacher teaches a particular lesson, as well as guiding the post teaching reflections to improve the lesson [15]. Lesson planning is pivotal to any useful teaching strategy and it provides a clear guideline as to what a teacher can do in the classroom [30]. According to Koning et al. [31],

there is evidence suggesting that lesson planning has largely depended on theories rather on empirical research. During the planning of a lesson, the teacher activates different knowledge bases for a particular topic to transform and render it teachable. An understanding of how in-service and pre-service science teachers activate the enacted PCK for planning would allow researchers to gather information on the nature of their activated knowledge and pinpoint the gaps in their PCK development. Therefore, we also explored in-service and pre-service teachers' enacted PCK with respect to planning to teach the PNM at junior secondary school.

## 3. Theoretical Framework

The quality of teacher knowledge is important for effective teaching and learning [2]. According to Shulman [1], good teachers have PCK, which he referred to as a "blending of content and pedagogy into an understanding of how particular topics, problems, or issues are organized, represented . . . ". He contends that this amalgam is adapted to the diverse interests and abilities of learners and presented as instruction to enhance learning. Ongoing work on teacher knowledge continues to advance the need to refine understanding about the nature of PCK components and their interaction in transforming a specific topic [6,24]. This endeavour is exemplified by the work of Krepf et al. [9], who analysed video-taped lessons on solubility concepts based on the integration of two PCK components, content knowledge (CK) and pedagogical knowledge (PK), as components suggested by Shulman [1]. They identified significant differences in the teacher knowledge components activated by expert and novice chemistry teachers. They maintained that the experts integrated and interconnected more knowledge components when transforming the content of the topic in ways which translated to effective teaching. Another perspective of PCK component interaction is portrayed in the work of Park and Oliver [32], who organised five components of PCK in their Pentagon Model to include orientations to science teaching, knowledge of students' understanding in science, knowledge of science curriculum, knowledge of instructional strategies, representations for teaching science, and knowledge of assessments of science learning.

Other research on teacher knowledge is the recently established refined consensus model (RCM) of PCK [6], which conceptualised three types of PCK, namely, collective (cPCK), personal (pPCK), and enacted (ePCK), within the grain size of a specific discipline, topic, or concept. The collective PCK is the specialised common knowledge generated collaboratively by a group of expert teachers in a community of practice [6]. It depicts the common knowledge that is found, for instance, in textbooks and reference materials such as the expert content representation on the PNM developed by Loughran, Berry, and Mulhall [33], which was available to the teachers in this study. Personal PCK is considered the private specialised knowledge held by an individual and developed through engaging with collective PCK and other experiences in practice. The enacted PCK is formed when personal PCK is operationalised in planning, delivery, and post-teaching reflection on a specific lesson [6].

This study explored the enacted PCK in planning and was conceptualised by drawing on a hybrid framework [34]. The framework is a blend of ideas from the refined consensus model of PCK [6] where the enacted PCK of this study is rooted, and the topic-specific PCK construct has five components [5] through which transformation of the topic concepts occurs. Specifically, the five components informed the design of a previously validated rubric that was used in this study to assess the components interactions attained in operationalising personal PCK that manifested as enacted PCK in planning. This hybrid framework highlights and makes more explicit the interrelated pathways through which a teacher can draw on their understandings of the different PCK components in constructing PCK within the collective PCK, personal PCK, and enacted PCK realms discussed earlier (see the model on the right-hand side of Figure 1). In another point of relevance, the framework clearly portrays reasoning through the five topic-specific components, presumably interchangeably through the three realms of PCK, where in the current context, enacted

PCK is linked with transformation of the concepts of the PNM during planning (see the model on the left-hand side of Figure 1).

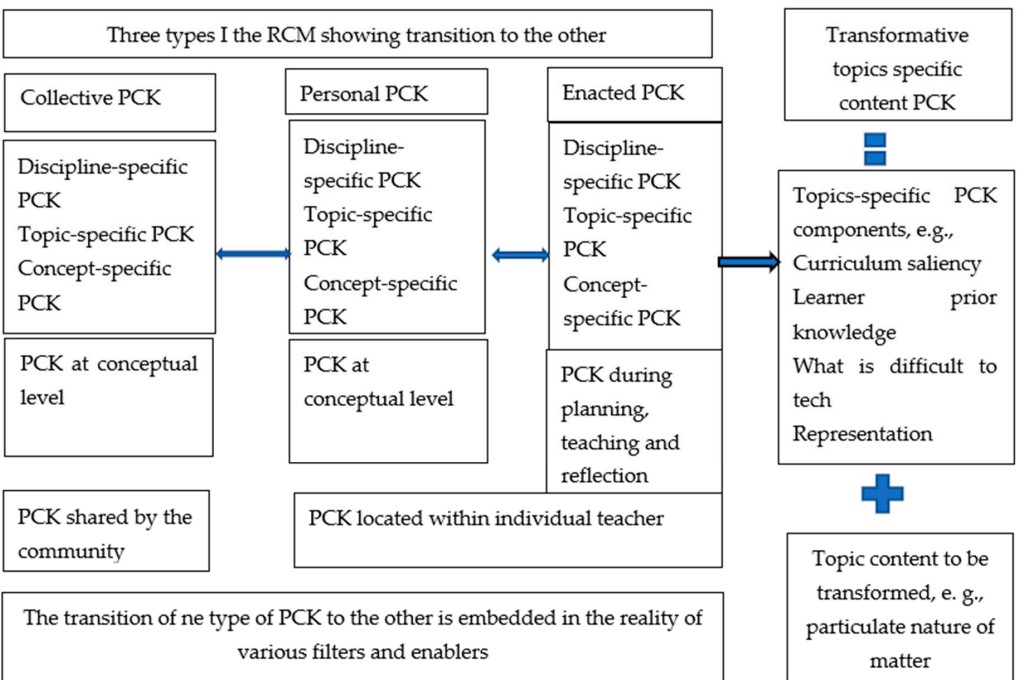

**Figure 1.** Hybrid PCK framework. Note: Hybrid PCK model showing the types of PCK in the refined consensus model (RCM) of PCK and the topic-specific PCK components adapted from Miheso and Mavhunga (2020).

The five topic-specific PCK components are as follows. The first is curricular saliency (CS), which includes the teacher's understanding of the important concepts and sequencing of concepts of the topic. The next is learner prior knowledge (LPK), which includes the prior knowledge and everyday ideas that learners bring to the classroom about the scientific content. The third component is knowledge about the use of representations (REP), which encompasses suitable examples, explanations, analogies, and representations at macroscopic, sub-microscopic, and symbolic levels. The next component is knowledge of what makes a topic difficult to teach and learn (WDL), which refers to potential areas that may be difficult for learners to understand, for example, understanding the existence of empty spaces between particles in a substance. The last is the conceptual teaching strategies (CTS), which refer to a teacher's knowledge of how to combine the other four components interactively into a suitable teaching strategy in a specific classroom context. In this study, for example, a teacher could plan to engage the students through a class discussion, whereby an appropriate example, diagram, or real object is used to enhance and complement the explanation of a certain concept, thereby addressing any specific learner misconceptions and developing learner understanding. In this case, more than one component is used interactively to transform the concept, thus portraying the teacher's enacted PCK in planning as being of superior quality. Using one PCK component alone cannot be considered a rich and effective PCK.

## 4. Research Methodology

In this study, we focused on capturing the type and quality of topic-specific PCK enacted during planning by practising and pre-service teachers. This study reports on explanatory sequential mixed methods research, where quantitative data were collected and collaborated with qualitative data [35]. The study was conducted among practising and pre-service teachers who were willing to participate in the study. The sample consists of 22 in-service or practising science and natural science (NS) teachers from eight high

schools around Pataka (pseudonym) and 22 pre-service teachers who were drawn from a cohort of 35 pre-service teachers enrolled in the Postgraduate Certificate in Education (PGCE) in NS teaching at a university in Njariani province (pseudonym). The PGCE course is run over two semesters and in each semester, students attend two double lectures a week and further complete 10 weeks of school experience. Participating pre-service teachers in this study would have observed experienced teachers teach the PNM topic. They were required to complete written assignments, lesson planning, microteaching, and peer teaching. According to Anderson and Mitchener [36], a science methods course helps pre-service teachers to integrate knowledge and gain experience in applying this knowledge during practice teaching.

## 5. Data Collection

Data collection involved a completion of a TSPCK questionnaire [12] by each of the participant teachers, who were instructed to do so within one and a half hours. The teachers' biographical information was collected at the same time. An excerpt of the instrument that related to teacher understanding in addressing alternative conceptions and misconceptions as elements of the learner prior knowledge component of TSPCK is presented in Figure 2.

---

**CATEGORY A: LEARNERS' PRIOR KNOWLEDGE**

1. Learners in a Grade 8 class were asked to represent the change that takes place when a substance is heated. The response Thabo has written on the board is shown below.

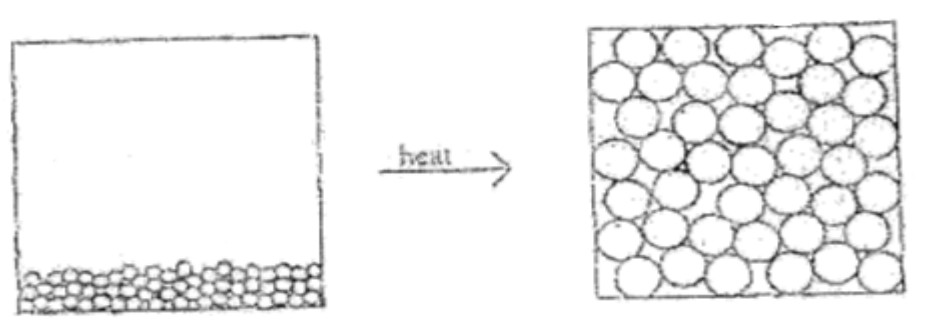

How would you comment on Thabo's response as part of an explanation to the rest of the class?

Note: An excerpt of the instrument related to teachers understanding and addressing learner misconceptions.

---

**Figure 2.** Excerpt from learners knowledge test item.

For the in-service teachers, the questionnaire was administered at their various schools, while the pre-service teachers completed the questionnaire at the university after completing four weeks of practice teaching in schools. The in-service teachers were recruited by the lead author at their various schools who had a brief session with them for an hour after school at each of the schools to explain the purpose of the study and before administering the questionnaire. In the case of the pre-service teachers, the questionnaire was administered by the second author, who was the lecturer of their course. The questionnaire was divided into five dimensions according to the five TSPCK components discussed previously to measure PCK at a topic level [37]. Each category had two to three questions, which were teacher tasks designed to elicit and capture teachers' PCK.

## 6. Data Analysis

The profile of the teachers was established, followed by the analysis of quantitative and qualitative aspects of the data concerning TSPCK.

### 6.1. Profile of Participants

Among the in-service teachers, the years of teaching experience varied between 1 and 30 years, with an average of 17. In terms of gender, 15 (68%) were female and 7 (32%) were male. In the case of pre-service teachers, 17 (77%) were female and 5 (23%) were male. The practising teachers held a wide range of teaching qualifications, from a Diploma in Education (Dip.Ed), Advanced Certificate in Education (ACE), Bachelor of Education (B.Ed.), B.Ed. (Hons.), through to a Master's in Education (M.Ed.). They nevertheless had similar backgrounds regarding teaching subjects, with 18 (82%) having at least one major in either chemistry, biology, or physics.

### 6.2. Quantitative and Qualitative Data Analysis

The quantitative analysis involved the coding of responses by assigning numerical values that were captured as limited (1), basic (2), developing (3), or exemplary (4). To test the validity and reliability of the tool used for the quantitative data, the scores were then subjected to the Rasch Unidimensional Measurement Model (RUMM 2030) (a computer software for analysis) to establish whether the items and persons would fit the model expectation of RUMM 2030, and to find out if there was any statistically significant difference between the means of the practising and pre-service teachers, an endeavour that was considered to be the quantitative analysis.

Qualitative analysis was performed deductively through the content analysis method [38]. It involved carefully reading through the respondents' comments for each of the test items, and then emerging themes were identified based on the levels of integrational TSPCK components. Further qualitative analysis of four case study teachers (two from each group, practising and pre-service teachers) was carried out to capture the conceptual teaching strategies and the learner prior knowledge test items, as these two components are reported to be common on the item map of PCK analysis [39]. This analysis allowed a deeper case-oriented analysis to give results in accordance with the ultimate reason for the qualitative inquiry. The selection of the four cases was based on the researchers' viewpoint [40,41] with the aim of providing insightful and comprehensive information regarding the research question, thereby facilitating insight into the teachers' enacted PCK that is incorporated in planning.

Both authors used the rubric for the conceptual teaching strategies test item (Table 1) in conjunction with the TSPCK components to deductively code each comment independently. Each TSPCK episode was identified and assigned the appropriate component and quality level relating to evidence of enactment of two and more of the five topic-specific PCK components in an interconnected manner in the episode. For example, in responding to a test item, a practising or a pre-service teacher responded to a conceptual teaching strategies test item and commented how, in transforming the content, they would discuss or explain a certain concept. For instance, a teacher might mention that particles in liquids are close together and are able to move randomly, slide past each other, and flow slowly in different directions (curricular saliency), and in the process also plan to use a representation (such as zigzag drawing to illustrate the idea) (representation) in addressing a possible learning difficulty relating to the abstractness of the topic content (what is difficult to teach) and learner challenges in understanding the arrangement, spaces, and constant movement of the tiny particles in substances (learner prior knowledge). Such a response involves an explanation with four components (curricular saliency, reproduction, what is difficult to teach, and learner prior knowledge) which are used interactively in transforming a specific topic content such that the pedagogical competence becomes integrated with other components (conceptual teaching strategies) and would be coded as an exemplary quality level of four out of the five components indicated by the rubric. This analysis exposed the type

and depth of quality of the enacted PCK in planning to transform the content of the PNM when planning, as revealed by the participants. Prior to the analysis, inter-rated reliability had been established. For this process, five samples of the completed TSPCK test were isolated and rated by the two authors and a chemistry expert; the various scores (especially where there was disagreement) were discussed for peer validation ensuring accuracy of the content and confirming the reliability of scores. The agreement was calculated using Cohen's Kappa inter-rater reliability, and the inter-rater agreement indicated an acceptable range of 0.75 and 0.77 for the practising and pre-service teachers, respectively. Possible researcher bias was further countered by rigidly adhering to the category descriptors in the rubric and by robust discussion of the results that were attained from independent coding and analysis by the two authors before comparing for agreement with respect to the research question. The initial findings were presented at a science education symposium, and comments from the audience helped to shape the final output.

**Table 1.** Excerpts from the scoring rubric.

| Limited (1) | Basic (2) | Developing (3) | Exemplary (4) |
| --- | --- | --- | --- |
| No evidence of acknowledgement of student prior knowledge and misconceptions Lacks aspects of curriculum saliency Uses representations limited to microscopic or symbolic scientific symbolic representation | Acknowledges student misconceptions verbally with no corresponding confrontations strategy Lacks aspects of curriculum saliency Uses microscopic and symbolic representations with no linking explanatory notes | Confirms and confronts students prior knowledge, misconceptions Uses at least one aspect related to curriculum saliency, sequencing or what not to discuss yet or emphasis of important concepts Uses at least two different levels of representations to enforce understanding | Considers students prior knowledge and evidence of confrontations of misconceptions Uses at least two aspects related to curriculum saliency, sequencing or what not to discuss yet or emphasis of important concepts Uses either microscopic or symbolic representations with sub-macroscopics representations to reinforce understanding |

Note: An excerpt from the rubric showing categories for coding conceptual teaching strategies test items from Pitjeng-Mosabala and Rollnick (2018).

## 7. Findings and Discussion

### 7.1. Differences in the Enacted PCK Activated by Pre-Service and in-Service Teachers

The results show a statistically significant difference in the understanding of the TSPCK component by pre-service and in-service teachers ($p < 0.006$). The pre-service teachers had a better understanding and showed a stronger interrelated integration of the five eTSPCK components.

The results of the quantitative analysis from RUMM 2030 revealed that the measures for individual-item and person-fit residuals displayed scores of $-2.5$ to $2.5$, which are well inside the acceptable fit, thereby indicating good validity. The Cronbach's Alpha measure of 0.75 revealed that there was reliability between the test items, because the nearer this score is to 1, the greater the reliability. These analyses thus indicate that all items behaved as expected by the model and so are a reliable measure of PCK for transforming the PNM in planning. This led to an answer to the second research question where a further analysis was conducted to compare the two groups. This revealed that, as described earlier, although both groups of teachers had similar backgrounds in their teaching subjects, their scores revealed that for the practising teachers, the mean of 0.67 and standard deviation (SD) of

0.40 were higher than for the pre-service teachers, with a mean of 0.65 and SD of 0.38. The ANOVA test results are presented in Table 2.

The one-way ANOVA results in Table 2 show that there is a statistically significant difference, F (1.20) = 9.522, *p* = 0.006, between in-service and pre-service teachers' responses to the various aspects of TSPCK of the PNM. These results show that the pre-service and in-service teachers understood the items on PNM differently.

With the scale set at zero logit or 0 item location, the higher values or the higher the item location on the scale in relation to the person score indicate that the test item was more difficult, and vice versa. A combined person item map for both groups of teachers was constructed to illustrate the items that posed some difficulties to both groups of teachers. The results in Figure 3 show that the in-service teachers found five items, such as what is difficult to teach (WDTI), conceptual teaching strategies, (CTSI), learner prior knowledge (LKI1), representation (RE1I) and (RE2I), and curricular saliency (CS2I) to be difficult (positioned above the 0-item location). These results also show that the pre-service teachers experienced the test items as less difficult than did the practising teachers (all the items below the 0-logit). The results are expected, since the PGCE programme promotes good teaching and PCK development. This presents a positive response to the second research question, and the subsequent in-depth discussion of the theme illustrates the nature of enacted PCK in planning among the pre-service teachers.

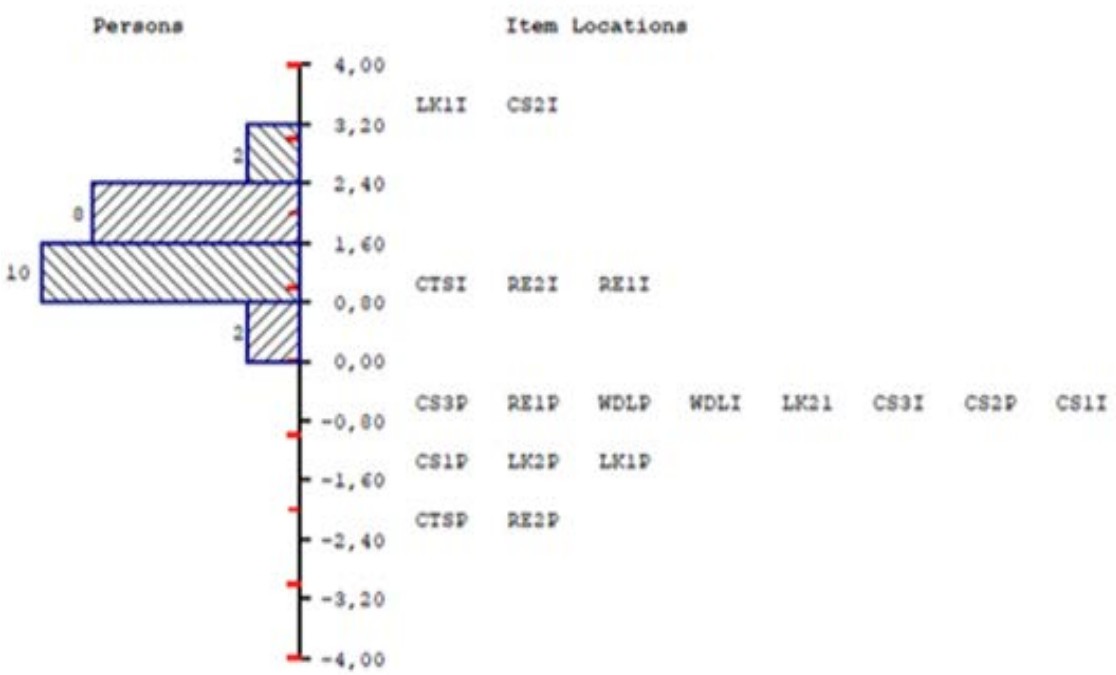

**Figure 3.** Item map. Note: An item map showing performance per TSPCK test component for the in-service (I) and pre-service (P) teachers.

**Table 2.** ANOVA test results.

| One-Way Analysis of Variance | | | | | |
| --- | --- | --- | --- | --- | --- |
| Source | Sum of Square | df | Mean Square | F | Probability |
| Between | 6.26 | 1 | 6.26 | 9522 | 0.006 |
| Within | 13.16 | 20 | 0.66 | | |
| Total | 19.42 | 21 | | | |

Note: One-way analysis of variance comparison of in-service and preservice teachers' TSPCK on the particulate nature of matter.

### 7.2. Pre-Service Teachers' Understanding and Integration of the Five eTSPCK Components

An in-depth qualitative analysis was carried out of the four case study teachers, referred to as pre-service teacher 1 (P1), pre-service teacher 2 (P2), in-service teacher 1 (I1), and in-service teacher 2 (I2). The analysis revealed several instances of TSPCK episodes portraying the activation of topic-specific teacher knowledge based on the five components through which the enacted PCK of the teachers was investigated (for type). It also showed the natural interplay between the components to signify quality with reference to the conceptual teaching strategies and learner prior knowledge test items. The test item required the teachers to describe how they would teach the lesson on PNM in the gaseous phase taking into consideration the other four TSPCK components and learner responses that include both correct understandings and misconceptions inherent in the test item, as indicated in Figure 4.

The learner prior knowledge test item (Figure 2) required the participants to describe how they would address learner misconception expressed in a learner response by providing an explanation to the rest of the class that reveals understanding and integration of the different knowledge components to correct the misconception and enhance learning.

The diagram below shows four learners' responses, A, B, C and D

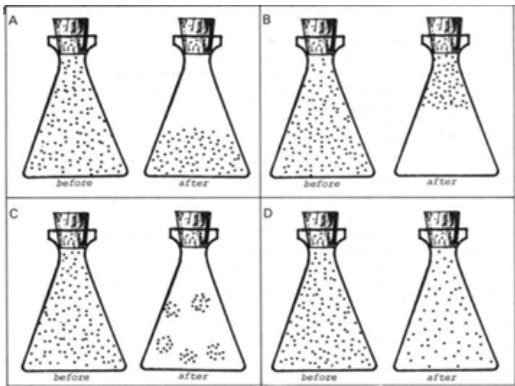

Following these four learners' responses, how will you teach a lesson on PNM in the gaseous phase? Explain fully showing any analogies or representations you would use and how you would deal with their response.

**Figure 4.** Question on conceptual teaching strategies. Note: Question on CTS relates to teachers' understanding of how to teach the lesson on PNM in the gaseous phase taking into consideration the other TSPCK components and learner responses that include both correct understanding and misconceptions.

The results in Table 3 show the type and quality of TSPCK components activated by the teachers about PNM. The finding indicates that pre-service teachers (P1 and P2) integrated the TSPCK components in planning better than the practising teachers (I1 and I2). Accordingly, Tables 4 and 5 summarise findings from the additional analysis on the conceptual teaching strategies and learner prior knowledge components. The figures further illustrate the better conceptual teaching strategies and learner prior knowledge with inherent component interaction for PCK quality of the pre-service teachers compared to the in-service teachers. This phenomenon is less frequently observed if the emphasis is based on the activation of only one component at a time in transforming the topic in question, meaning that the entwined logical use of more components indicates superior enacted PCK. The names of the topic-specific PCK components are abbreviated for ease of reference as follows: CS, curricular saliency; LPK, learner prior knowledge; WDL, what is difficult to teach; REP, representations; CTS, conceptual teaching strategies. To elaborate on the

activated knowledge and portray the knowledge component interaction that characterised the evidence as summarised in Tables 4 and 5 for CTS and LPK, respectively, extracts of the teachers' responses to both test items are given in Tables 4 and 5 as sub-themes to the second theme of pre-service teachers' superior understanding and interrelated integration of the five eTSPCK components.

It is important to highlight the idea that in this study, it is assumed that the collective PCK refers to relevant teacher knowledge about transforming a specific topic content developed by expert teachers from shared insights. This knowledge is made available in reference materials and teacher resources to would-be and existing teachers for consultation and to draw on to inform their personal PCK that, in turn, informs their enacted PCK in the form of integrated knowledge components to enhance transformation of concepts in a topic. In this study, pre-service teachers accessed the collective PCK by virtue of participating in the programme and engaging with its inherent curriculum, or through personal efforts to source additional reference materials to enhance their understanding of the would-be practice. The latter is true for in-service teachers with the awareness for continuously improving their practice. Therefore, accrued understandings and interaction of knowledge components revealed as enacted PCK in planning manifested by the two groups of case study teachers in their responses to the conceptual teaching strategies and learner prior knowledge test items are entrenched in the six sub-themes discussed below.

**Table 3.** Summary of activated TSPCK components by case study teachers.

| Participant | CS | REP | LPK | WDT | CTS |
|---|---|---|---|---|---|
| P1 | Developing | Developing | Exemplary | Developing | Exemplary |
| P2 | Developing | Developing | Basic | Developing | Developing |
| I1 | Basic | Basic | Basic | Basic | Developing |
| I2 | Basic | Basic | Basic | Limited | Basic |

Note: Summary of case study participants' type and quality of activated TSPCK components.

**Table 4.** Summary of scores and integration of TSPCK components for CTS test item.

| Participants | CTS scores | Other TSPCK components integrated with the CTS |
|---|---|---|
| P1 | Exemplary | LPK, REP, WDT, CS, and CTS |
| P2 | Developing | CS, REP, and LPK |
| I1 | Developing | CS, REP, and LPK |
| I2 | Basic | CS and REP (Ineffective) |

Note: Summary of the case study participants' scores test item and interconnected use of the other components.

**Table 5.** Summary of scores and integration of TSPCK components for LPK test item.

| Participants | CTS scores | Other TSPCK components activated in relation to LPK test item |
|---|---|---|
| P1 | Exemplary | REP, CS, LPK, and CTS |
| P2 | Developing | LPK and REP |
| I1 | Developing | LPK and CS |
| I2 | Basic | LPK and CS |

Note: Summary of the case study participants' scores on the LPK test item and interconnected use of the other components.

## 8. LPK, REP, WDT, and CS Linked to Conceptual Teaching Strategies Test Item as Exemplary Enacted PCK Activated by Pre-Service Teacher 1 (P1)

The evidence shown in Figure 5 indicates that the response to the conceptual teaching strategies test item of the knowledge activated by pre-service P1 entailed drawing from the collective PCK presented at the teacher preparation programme. Accordingly, this informed her personal PCK, which she activated and drew from to construct and enact her PCK for transforming the PNM in the gaseous phase while focusing on addressing a learner misconception.

In revealing the interconnected and complementary use of the different enacted PCK knowledge components in planning during a TSPCK episode, pre-service teacher P1 draws strategically from her personal PCK. She does so to initiate progress in transforming the concepts by requesting learners to explain using an appropriate diagram of the flask (portrayed in the diagram in Figure 5) (LPK) depicting a sub-microscopic representation (REP) of gas particles that move randomly, filling up the whole space available in the container (CS) and in the process corrects the misconception (LPK). Teacher P1 also planned to use a video simulating the actual movement of the gas particles, which is another sub-microscopic representation (REP), to explain and emphasise the standard scientific understanding of the concept. Representing the same concept in three different ways (the words in the explanation, a symbolic representation and the appropriate diagram and video of both sub-microscopic representations) in a lesson signifies drawing from the teacher's deep ability and reasoning in transforming the topic content when planning. This concurs with Krepf et al. [9], who found that teachers with great insights about teacher knowledge for teaching can activate complex combinations when enacting their lesson. This pedagogical approach has the potential to help students understand the molecular representation of matter, thus addressing a problem that has been reported in the literature [11,42]. The teacher's response also includes knowledge of what aspect of the topic learners would find challenging (WDL). Based on the preceding information, P1 drew on five different TSPCK components in an interactive and complementary manner to transform the concept, thus providing evidence of superior quality enacted PCK that was scored as exemplary.

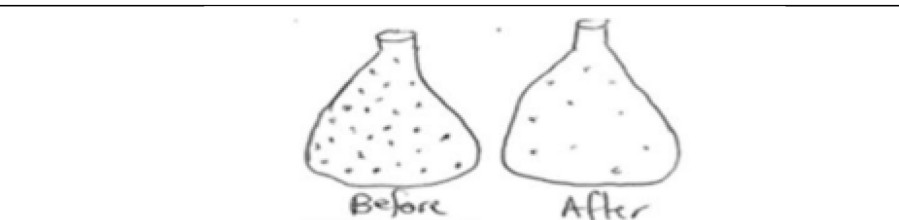

I would use these analogies to ask questions to the learners on how they think particles in a gas phase are arranged. The learners would have to explain using one of the images to show what happens after a gas is removed by the pump from the flask. I would look for answers such as gas particles have no particular arrangement and move at random motion and move more freely to fill up the entire space of the container. They do not clump up anywhere. I will also use a video and explain to demonstrate the actual arrangement and random movement of particles which learners find difficult to understand.

**Figure 5.** Except from responses to CTS test by P1. Note: An excerpt from response to CTS test item by PI that revealed effective integration of LPK, REP, WDT, and CS.

### 9. CS, REP, and LPK Linked to Conceptual Teaching Strategies Test Item as Developing Enacted PCK Activated by Pre-Service Teacher 2 (P2)

The response of P2 to CTS, shown in Figure 6 below, revealed activation of CS, REP, and LPK as enacted PCK.

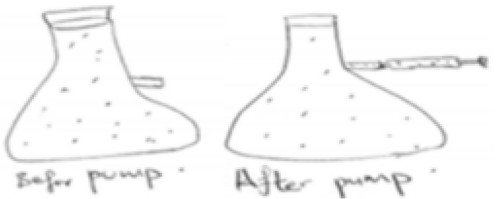

I love using videos. They are captivating and can aid for visual learners as well.. I would also try to make coloured smoke bombs sold at the shop and are for use. Therefore, I would use the diagrams above to explain that when gas moves from one container to another it does not form groups at certain points. But rather move from high concentration to region of low concentration. It will not accumulate at a certain point because gas particles have weak forces between them, they spread out and eventually occupy the entire container.

**Figure 6.** Excerpt from response to CTS test item by P2. Note: An excerpt from response to CTS test item by P2 that revealed effective integration of LPK, REP, WDT, and CS.

In the above excerpt in Figure 6, P2, like P1, is strategic in drawing from his personal PCK to activate his enacted PCK in planning. His strategy includes knowledge of representations that involved showing a video of gas particles, diagrams (sub-microscopic REP), and using a coloured smoke bomb (macroscopic REP) to support the explanation of the constant, random movement of gas particles to fill up the whole container uniformly (CS). The enacted PCK thus portrayed simultaneous interaction of the knowledge components in transforming the concept with the aim of addressing the learner misconception that gas particles accumulate when some particles are removed from the container (LPK). However, he achieves this without integrating what is a difficult to teach component. Accordingly, his enacted PCK in planning was scored as developing.

**10. CS, REP and LPK Are Linked to Conceptual Teaching Strategies Test Item as Developing Enacted PCK Activated by In-Service Teacher 1(I1)**

The excerpt below in Figure 7 reveals how teacher I1 integrated the CS, REP, and LPK knowledge components.

What leaners need to know about particles in a gaseous state is that particles are free to move, and they are widely packed. An example of a gaseous phase is 20 students moving in a lecture hall, so it can be explained to learners that if more students enter the class, movement of the first 20 will not be easier if a class becomes completely filled. What learners need to know is that if a gas is removed in a flask, movement of the particles will be free, but there will be a decrease in the number of particles left.

**Figure 7.** Excerpt from responses to CTS test item by I1. Note: I1's comment that reveals integrated understanding of CS, LPK and REP.

As shown in the except above, teacher I1 drew from his personal PCK in transforming the content when planning, through his emphasis that gas particles move freely (CS) before incorporating a practical analogy in which students simulate the gas particles moving freely into the lecture hall; this is an example of using a sub-microscopic level of representation (REP) for the concept of free movement of gas particles. His planned use of the analogy, without clearly identifying the misconception as required in the rubric, nevertheless by implication supports his confronting the student misconception (LPK) because he states that gas particles will continue to move freely to occupy the whole container even when some of the particles are removed (CS). From the preceding discussion, although I1 integrated three

forms of knowledge components (REP, LPK, and CS) as enacted PCK, the response does not specifically address the student misconception and only one level of representation is integrated, which led to his enacted PCK being rated as developing.

### 11. CS and REP as Basic Enacted PCK Activated by in-Service Teacher 2 (I2) in Relation to Conceptual Teaching Strategies Test Item

Another example to illustrate the knowledge integration by practising teachers is the case of I2, who activated CS and REP, as reflected in the excerpt shown in Figure 8.

Gas spread out quickly to fill the space available. Gas particles move very fast, much faster than in solids and liquids. The particles in gas possess a lot of energy. Have you ever tried to compress the gas in a syringe or in a bicycle pump? Why do you think you can compress a gas? In a gas the forces between particles are very weak. This explains why the particles in gases are not neatly arranged. They have large spaces between them. These are much larger than in a solid and liquid state.

**Figure 8.** Excerpt from responses to CTS test item by I2. Note: I2's and I1'scomment that reveals integrated knowledge of CS and REP (but ineffective).

It is clear from the above excerpt that, in operationalising his personal PCK during planning, in-service teacher I2 explained that gas particles have high kinetic energy and weak forces between them and so move rapidly in different directions; these are all important conceptions associated with CS about the PNM. The understanding is incorporated in conjunction with knowledge of representation (REP) in the example of compressing air in a syringe or bicycle pump. This macroscopic level of representation of gases, although indicating enacted PCK with interconnected integration of knowledge of representation and curricular saliency, is ineffective in the given context. It does not enhance students' conceptual understanding at the sub-microscopic level of representation of particles; this had been identified as problematic in the literature [42] and so was incorporated specifically in the conceptual teaching strategies test item. Consequently, his enacted PCK on CTS component was rated as only basic.

### 12. REP, CS, LPK, and CTS as Exemplary Enacted PCK Activated in Relation to Learner Prior Knowledge Test Item by Pre-Service Teacher 1 (P1)

The evidence shown in Figure 9 indicates the superior teacher knowledge activated by pre-service P1, as she drew from four components interactively in transforming the content relating to heating of a substance while focusing on addressing a learner misconception.

I will refer to the diagram and draw a correct one. I will explain to the class that molecules do not change in size when heated. When heated molecules gain kinetic energy and start vibrating and the bonds between them weaken to allow re-arrangement and in some cases allowing the molecules to move apart from each other and space increase between them. It is difficult to see this movement with a naked eye. I will use animated videos for learners to observe what happens to the particles when a substance is heated and ask questions. I will demonstrate with an example of boiling water in a see trough kettle and ask questions to make suer tat learners understand.

**Figure 9.** Excerpt from response to LPK test item by P1. Note: P1's comment that reveals a superior integrated knowledge of 4 components, REP, CS, LPK, and CTS.

As the response in Figure 9 implies, P1 revealed knowledge of misconception that particles increase in size by drawing from her personal PCK knowledge base of common misconceptions (LPK) about the topic. She expanded by drawing on her knowledge of the curricular saliency in explaining that it is the space between the particles that increase as the substance heats up and molecules rearrange (CS). Teacher P1 also indicates the use of a diagram as well as video simulating the actual movement of particles, which are sub-microscopic representations (REP). The plan to use macroscopic representation of water to explain and emphasise the standardised understanding with heating of a substance portrays a significant understanding and knowledge of integration of representation with two other components of TSPCK, interactively culminating to conceptual teaching strategies in addressing the learner prior knowledge test item. Accordingly, the preceding understanding and integration of knowledge components was rated as exemplary quality of enacted PCK that manifested in planning to address expressed learner misconception.

### 13. LPK and CS as Basic Enacted PCK Activated in Relation to Learner Prior Knowledge Test Item by in-Service Teachers 1 (I1) and 2 (I2)

Another example to illustrate the basic enacted PCK integration by in-service teachers is the evidence portrayed in the case of teachers I1 and I2, who activated a basic enacted PCK of two components (LPK and CS), as reflected in the excerpt shown in Figure 10.

It is clear from the above excerpts that, in explaining how to address learner misconception perceived from a learner's response in the learner prior knowledge test item, both in-service teachers drew from their understanding of learner prior knowledge to identify the misconception that particles expand (LPK). They also drew from their knowledge of curricular saliency in integrating standardised knowledge that all substances have small particles called atoms that may combine chemically, and when heated, the space between the particles becomes bigger (CS from in-service case study teacher 1), and when heated, solids may become liquids and liquids may become gases (CS from in-service case study teacher 2). Considering that they integrated LPK and CS and there was no evidence of drawing from the other knowledge components, a more direct teacher talk instructional approach with a simple quality of enacted PCK was indicated and was rated as basic.

> I will explain that all substances have small particles called atoms that may combine chemically with each other to form molecules. I will explain that when substances are heated their molecules do not expand in size but the space increases. I will ask learners to explain. (Excerpt from response of I1.)
>
> I will explain that when a substance is heated, the molecules become freer from each as the bonds are weakened; in some cases bonds break completely and the space between particles becomes bigger. If this happens, substances that were originally solids may become liquids and liquids become gases. (Excerpt from response of I2.)

**Figure 10.** Excerpt from response to LPK test items by in-service teachers I1 and I2. Note: Comments from I1 and I2 that revealed basic integrated knowledge of two components each.

### 14. Discussion and Conclusions

Overall, the findings of this study converge on the point that the integration of the components into quality PCK in transforming a concept is not a straightforward process that depends mainly on the simple possession of those components. Rather, the quality of the PCK is influenced by strategic integration of different knowledge components [24,34]. The results shown in the two themes and six sub-themes discussed above have contributed to our understanding of practising and pre-service teachers' enacted PCK in planning with respect to the PNM. This is due to the investigation and comparison of the type and quality of enacted PCK constructed by these two groups of teachers as they drew

from their personal PCK informed by the collective PCK in the context of planning when transforming topic-specific content. Evidence from Rasch analysis, RUMM, revealed a statistically significant difference between the type of knowledge activated by the two groups of teachers ($p < 0.006$). The pre-service teachers found the test items to be less difficult than did the practising teachers. This result is unexpected since PCK is known to improve with practice. The findings of the current study contradict Krepf et al.'s [9] study on analysis of a lesson on optics and reported that practising teachers activated significantly more knowledge components than did the novice teachers. The findings from this study can be explained by the context and experiences of pre-service teachers in their PGCE course programme, which concerns effective teaching, and the TSPCK components are integrated in the course. The in-service teachers in this study derived from their experiences with limited support from policy documents on the TSPCK component.

The other difference could be due to language; in this case, the pre-service teachers were able to articulate their ideas in writing. Regarding the type and quality of ePCK activated by the teachers when drawing from their personal PCK and the collective PCK in transforming the topic concepts, pre-service teachers (P1 and P2) both activated CTS, CS, REP, and LPK, but pre-service teacher P1 had an additional component, WDT. They used the components in a complementary and interactive manner. The quality of enacted PCK for the two pre-service teachers was thus found to be exemplary and developing in the conceptual teaching strategies test item. In the case of the two in-service teachers (I1 and I2), they both integrated the CS and REP components, and also LPK activated by practising teacher I1; however, the knowledge of REP activated by I2 was not strong in identifying student misconception about rapid random movement of gas particles to fill the entire container, in that it was only a macroscopic representation. As such, it could have been more appropriate to use it in combination with sub-microscopic levels of representation to combat the misconception. The quality of the two practising teachers' enacted PCK was found to be developing and basic, respectively (see Figure 9). Although the knowledge components activated by all four teachers differed, the pre-service teachers activated more knowledge components and demonstrated a better understanding of the enacted PCK needed for transforming the concepts of the topic to enhance learning in the context of planning. Coetzee, Rollnick and Gaigher [43] reported that it was difficult to accomplish rich interaction among the components of topic-specific PCK when the participants felt intimidated by concepts or did not have a sound conceptual understanding of the concepts. The concept TSPCK is not necessarily a term commonly used by teachers to describe or explore understandings of their practice. This provides a rationale for the poorer performance shown in our study by the practising teachers. As mentioned, the pre-service teachers were enrolled in a programme which specifically incorporated TSPCK components. The practising teachers had qualified some years earlier and one can surmise that they had not been previously exposed to the topic and specific components, and there is thus promise in their contribution to our understanding of the teacher knowledge used by novice and practising teachers. This knowledge is important for informing and refining teacher development programmes, such as teacher education and continuous professional development interventions for teaching. The results of this study implicitly highlight the value of incorporating interactive use of the TSPCK components so teachers can understand the pedagogical practices pertaining to transforming specific topics and related concepts. Accordingly, we recommend its inclusion in teacher development programmes. This study has some limitations in that it focused on the enacted PCK aspect only in the context of planning in transforming the concepts in a topic. In addition, the sample size was small (less than 50 teachers), which means that the findings cannot be generalised. Irrespective of the shortcomings of this study so highlighted, Shulman's [1] notion that pedagogical reasoning is as important as the actual enactment of the lesson should be noted. This study has provided findings that should encourage and inform further investigations on pedagogical reasoning; therein lies its merit. Therefore, a further recommendation is to encourage future investigations based on interactive use of TSPCK components in understanding the nature

of teachers' knowledge used through the complete teaching cycle of planning, teaching, and post-teaching reflection to provide a more comprehensive account of a lesson.

**Author Contributions:** Conceptualization, A.B. and D.S.; methodology, A.B. and D.S.; software, D.S.; validation, A.B.; formal analysis, A.B. and D.S.; investigation, A.B. and D.S.; resources, A.B. and D.S.; writing—original draft preparation, A.B.; writing—review and editing, D.S.; project administration, A.B. and D.S. All authors have read and agreed to the published version of the manuscript.

**Funding:** This research received no external funding.

**Institutional Review Board Statement:** The study was conducted in accordance with the Declaration of Helsinki, and approved by the Ethics Committee of University of The Witwatersrand (protocol code 2014ECE014S and year of approval: 2014).

**Informed Consent Statement:** Informed consent was obtained from all subjects involved in the study.

**Data Availability Statement:** Not applicable.

**Acknowledgments:** We would like to say thank you to the participating teachers.

**Conflicts of Interest:** The authors declare no conflict of interest.

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
