# Peer review of "In-Service and Pre-Service Science Teachers’ Enacted Pedagogical Content Knowledge about the Particulate Nature of Matter"

_education, doi:10.3390/educsci12090576_

Round 1
Reviewer 1 Report
The manuscript aims at studying the enacted pedagogical content knowledge of practising and pre-service teachers. The study is interesting and the methods used appear robust and able to give meaningful answers to the research questions.
I would suggest to the authors to change the references to "students' misconceptions" (page 3, lines 104, 106, 109-110, 117) adapting them to the more modern idea of "students' common sense conceptions".
Author Response
Please see attached notes

Reviewer 2 Report
Thank you for the opportunity to review this interesting paper. The introduction had a good framing of PCK though a bit 'light' on references initially. The description of the 3 types of PCK beg a flow-chart or diagram to show their interconnection but this might duplicate or pre-empt Figure 1. Regarding the Figures, these should be redrawn so that the text matches the text style in the main paper. As it stands, the Figures look like a bunch of text boxes, and the use of a diagram drawing applet or app is recommended.
I note that Figures 2 and 12-15 are excerpts, but the full questionnaires could be placed in an appendix.
Please note the journal author guidelines which have been placed at the end of the paper. You need to include an informed consent statement, plus give details of the ethics requirements in order to carry out this study, and provide a data availability statement.
I was surprised by the paragraph on planning on page 4. Whereas this might have been self-evident, how strongly was planning used as an input parameter in this study. What was the relevance of including the Evens et al. (2017) citation on line 160ff? Its relevance was not explicit but implied.
Figure 4. does not comply to the standard style of reporting ANOVA results
Author Response
please see attached notes
